# Evolution of Medical Students’ Perception of the Patient’s Right to Privacy

**DOI:** 10.3390/ijerph191711067

**Published:** 2022-09-04

**Authors:** Alberto Álvarez Terán, Camilo Palazuelos, Trinidad Dierssen-Sotos, Jessica Alonso-Molero, Javier Llorca, Inés Gómez-Acebo

**Affiliations:** 1Faculty of Medicine, Universidad de Cantabria, 39011 Santander, Spain; 2IDIVAL Instituto de Investigación Sanitaria Valdecilla, 39011 Santander, Spain; 3Centro de Investigación Biomédica en Red Epidemiología y Salud Pública (CIBERESP), 28029 Madrid, Spain

**Keywords:** clinical practice, health policy teaching, medical students, patient consent, privacy

## Abstract

During clinical rotations, medical students experience situations in which the patients’ right to privacy may be violated. The aim of this study is to analyze medical students’ perception of clinical situations that affect patients’ right to privacy, and to look for the influential factors that may contribute to the infringement on their rights, such as the students’ age, sex, academic year or parents’ educational level. A cross-sectional study was conducted with a survey via “Google Drive”. It consisted of 16 questions about personal information, 24 questions about their experience when rotating and 21 questions about their opinion concerning several situations related to the right to privacy. A total of 129 medical students from various Spanish medical schools participated. Only 31% of 3rd–6th year students declared having signed a confidentiality agreement when starting their clinical practice, and most students (52%) reported that doctors “sometimes”, “rarely” or “never” introduce themselves and the students when entering the patients’ rooms. Additionally, about 50% of all students reported that they would take a picture of a patient’s hospitalization report without his/her (consent), which would be useful for an assignment. Important mistakes during medical students’ rotations have been observed, as well as a general lack of knowledge regarding patient’s right to privacy among Spanish medical students. Men and older students showed better knowledge of current legislation, as well as those whose parents were both university-educated and those in higher academic years.

## 1. Introduction

In terms of health, privacy is of notable importance, taking into account the situation of vulnerability in which the patient finds themselves, regarding pain, be it physical or not, or discrimination and social stigma [1]. Daily, health-care workers face situations that may violate patients’ right to privacy, not only concerning access to past medical records (previous illnesses, toxic habits, diagnostic procedures, physical examination), but also related to studies or research with images or detailed case descriptions where patients’ identity might not be safeguarded [2]. Furthermore, new information technologies have added a new danger by greatly facilitating access to vast amounts of health data.

In order to articulate their commitment to the advance and protection of human rights within their borders, many countries have incorporated a privacy rights protection scheme into their national constitutional and regulatory frameworks [3]. The World Medical Association refers to the patient’s right to privacy (among other rights). This was adopted by the 34th World Medical Assembly in Lisbon in 1982 [4]. Regarding Spain, its Constitution of 1978 refers to privacy in Article 18.1 as a fundamental right of the individual. In the same way, there are other deontological codes and national legislations about privacy protection, such as The General Health Law of 1986 on its Article 10.1, the Patient Autonomy Law 41/2002 of November 14, the Organic Law 1/1982 of May 5 on its Article 1.3, and the SSI/81/2017 Order of November 19 [5]. Privacy is so heavily guaranteed in the Spanish legal system that violations carry important economic fines, occupational prohibition and even prison for those professionals who violate their professional secrecy [6].

In this regard, it has been observed that there is a significant lack of knowledge among medical students concerning privacy and medical law in general [7]. Differences have been shown between what students think they are learning and what faculty professors think they are teaching in regard to ethical and legal issues [8]. Throughout the medical degree they experience different teaching methods, such as theoretical classes, lectures, seminars or clinical rotations. The latter, more frequent in the final academic years, has been reported to be useful in students’ effective learning [9,10], the amount of patients, the variety of clinical presentations, the engagement and involvement of the student’s supervisor and also the supervisor’s experience in clinical teaching being main factors [11]. During clinical practice, medical students participate in different health-care processes in which patients’ right to privacy may be violated. The main objective of this study is to analyze medical students’ perception of clinical situations that affect patients’ right to privacy or intimacy, and to look for factors that may influence this perception, such as the students’ age, sex, academic year or parents’ educational level.

## 2. Materials and Methods

### 2.1. Setting and Participants

A degree in Medicine lasts 6 years in Spain. Students usually rotate in a hospital from 3rd year on, but some medical schools could have earlier short periods of student–patient contact. Qualifications in Spanish universities are graded from 0 to 10. The required mark to pass any subject is 5. Students can progress to more advanced years without having passed each previous subject, although passing each and every subject is a requisite to graduate.

A cross-sectional study was conducted. The survey was designed for medical students in 1st to 6th year, and it was sent to 10 Spanish medical schools on 24 November 2019. The participants in the study were required to be studying a medical degree and be over 18 years of age at the time of agreeing to participate.

All the students who participated in this survey completed it online via “Google Drive”. The participants were contacted via e-mail and WhatsApp.

### 2.2. Gathered Information

The information was gathered through a web questionnaire, preserving the anonymity of the respondent. The survey included 61 questions which were classified into three groups: personal information questions, questions about students’ experience and questions was focused on students’ opinions regarding some situations where patients’ privacy may be violated.

Personal information questions included sex (male/female); whether the student’s parents are university-educated (later classified in both/one or none); the year of study the student is in (later classified in 1st to 3rd/4th to 6th); whether the student has subjects from previous years they still need to pass; the student’s current average mark (later classified in 5 to 6.9/7 to 8.9/9 or more/Prefer not to reveal); and the number of clinical services the student has rotated in (later classified in 0/1 to 9/10 or more) (Appendix A). Apart from 16 personal information questions such as age or gender (Appendix A), the main part of the questionnaire was composed of two different groups of questions concerning situations related to the SSI/81/2017 Spanish Order. Articles regarding this Order are shown in Appendix A.

The first group included 24 questions about students’ experience when rotating in hospital; these questions were only analyzed for students in 3rd–6th year, as clinical experience in the first two years of the Medicine degree is scarce or nonexistent. The aim of the first group of questions was to analyze the level of fulfillment of the current legislation among medicine students, by asking aspects about their experience in their clinical rotations. (Appendix A).

Twenty-one questions were asked about student’s opinion regarding several assertions and their attitude when facing some virtual situations related to patient’s privacy. These questions were analyzed for all respondents, whatever their year (Appendix A).

Answers to questions 1–45 in Appendix A were assigned a score from 0 to 1, where 1 refers to the answer considered the most adequate from the legal or ethical point of view.

### 2.3. Ethical Issues

This study was authorized by the Research Projects Ethics Committee of the University of Cantabria (CE TFG 11/2019). Before answering the questionnaire, the participants had access to an information sheet and an informed consent paper. Only after filling in and signing the consent form did the participants gain access to the questionnaire.

### 2.4. Statistical Analysis

Variables are described as frequencies and percentages or means and standard deviations.

First, to make all the questions comparable they were rescaled from 0 to 1, and some of them were reversed to be in the same direction. In other words, 1 point was assigned if the student answered according to compliance with the rules, and 0 points were assigned if he or she did not. Answers to questions 25–45 in Appendix A (i.e., those questions about students’ opinion on specific situations) were dichotomized as 1 = the student gave the answer considered correct—as indicated in Appendix A—or 0 = the student gave any other answer. Then, logistic regression was used to analyze the association between those answers and student-related factors, which could condition their knowledge about compliance with the regulations. In brief, usual logistic regression models the relationship between a variable exposure, say age, and the probability of knowledge about ethical issues, as:$$Log\left(\frac{P\left(y_{i}\right)}{1-P\left(y_{i}\right)}\right)=\beta_{0}+\beta_{1}age_{i}$$
where *P*(*y_i_*) is the probability that a student gives the answer that is considered correct for patient *i*; age, take values for each patient. In this formula, *β*_1_ is the natural logarithm of the odds ratio if the student gives the answer considered correct for each additional year in age.

In the same way, for the association between sex, academic year or parents’ educational level and knowledge about ethical issues, the following equations were used:$$Log\left(\frac{P\left(y_{i}\right)}{1-P\left(y_{i}\right)}\right)=\beta_{0}+\beta_{1}sex_{i}$$
$$Log\left(\frac{P\left(y_{i}\right)}{1-P\left(y_{i}\right)}\right)=\beta_{0}+\beta_{1}academic\;year_{i}$$
$$Log\left(\frac{P\left(y_{i}\right)}{1-P\left(y_{i}\right)}\right)=\beta_{0}+\beta_{1}parents^{\prime }educational\;level_{i}$$

The results are reported as odds ratios (OR) with 95% confidence intervals (CI). All reported *p*-values are two-tailed. All statistical analyses were performed with the Stata 16/SE software (Stata Co., College Station, TX, USA).

## 3. Results

The questionnaire was answered by 129 students. Descriptive data regarding the personal information of the students are shown in Table 1. The mean age was 22.1 years (standard deviation: 3.5). A total of 94 of the 129 respondents were female (73%) and 33 were male (26%), whereas two students did not specify their gender (1%), which is very similar to the current gender distribution of medical students in Spanish universities. About one third of the participants (33%) were in the first, second or third year of their medical studies. Two thirds (67%) were in fourth, fifth or sixth year, the latter being the one with the largest number of respondents (38).

### 3.1. Students’ Experience When Rotating in Hospital

Answers to questions about students’ experience when rotating in hospital are described in Table 2. Only 31% of 3rd–6th year students declared having signed a “confidentiality commitment” document when starting their clinical practice (question 2). Furthermore, there is a significant difference between the results of 3rd–5th year students and 6th year students. Only 18% of 3rd–5th year students declared having signed this “confidentiality commitment”, while the level of compliance among 6th year students was much higher (61%). Moreover, the huge majority of 3rd–6th year students (77%) declared that the doctors they were working with never provided them with their personal username and password to access the hospital’s internal network (question 3).

Most 3rd–6th year students (80%) reported that they “always” or “quite often” carry their identification card when rotating in the hospital (question 4). Only 2% of the students declared having returned their identification card when ending their rotations (question 6).

A lack of experience with dummies, roleplays and interactive dummies meant 75% of 3rd–6th year students had been nervous when treating a real patient at least once (question 10).

Regarding the frequency with which the doctor informs the patient of the presence of students during the care process (question 18 in Table 2), the most frequent answer was “quite often” (32%), but still, most of the students’ answers (57%) revealed that doctors “sometimes”, “rarely” or “never” notify the patient of their presence. In the same way, students declared having been introduced to the patient by the doctor “quite often” (36%), while most of the students’ answers (52%) reveal that doctors “sometimes”, “rarely” or “never” introduce the students and themselves when entering the patients’ rooms (question 22).

Private life issues of patients were reported to have been gossiped about by 58% of students at some point during their clinical rotations (question 20).

### 3.2. Students’ Opinion

Table 3 provides information about questions regarding students’ opinions on privacy issues. In general terms, most students’ answers were considered correct when they were asked about issues concerning medical secrecy and situations in which doctors are legally allowed to break it. A total of 91% of students knew they were subject to medical secrecy obligation (question 25), and 85% students were aware that medical secrecy remains after patients’ death (question 27). The results from questions 30 and 33 also follow this tendency, since students showed general knowledge (85%) about breaking medical secrecy in the cases of a disease of public health significance (DOPHS) and severe contagious diseases (68%).

Only 23% of students knew that doctors can legally break the patient’s medical secrecy if the patient gives their consent (question 26). About 50% of students knew that doctors are legally allowed to break the patient’s medical secrecy obligation if a judge calls them to testify in a trial concerning that patient (question 38). Question 31, regarding photographing a patient’s hospitalization report because it would be useful for an assignment, received quite diverse answers.

The students were asked about uploading a picture showing the auscultation of two patients to Instagram, having not asked for the patient’s consent in either case. The patient in question 39 had a characteristic tattoo on his sternum area, even though his face was not shown in the picture, while the patient in question 40 had no characteristic physical aspects and his face was not shown in the picture, so he could not be identified. About 94% students totally disagreed with taking and uploading the picture in question 39, but only 61% totally disagreed in question 40.

We analyzed all these results according to several variables to see if there were differences between various groups of students.

### 3.3. Association between Age and Knowledge about Ethical Issues

The influence of age on answers considered correct for questions 25 to 45 is shown in Figure 1.

In general terms, we found that older students tend to answer more correctly than younger ones. An example of this is question 34, about sending a picture of a case of Fournier’s gangrene via WhatsApp without the patient’s consent, where results indicate that the probability of answering this question correctly increases 1.30 times per year of age (95% CI 1.05–1.60, *p* = 0.017). The results from question 39 (OR = 1.75, 95% CI: 1.09–2.80, *p* = 0.021) and question 40 (OR = 1.25, 95% CI: 1.03–1.52, *p* = 0.023) follow the same tendency. In these questions’, the students were asked about uploading a picture auscultating two patients to Instagram, having not asked for the patient’s consent in either case.

### 3.4. Association between Sex and Knowledge about Ethical Issues

We compared the results between two groups according to the sex of the students (male or female). The results regarding this variable are shown in Figure 2.

In general terms, the results show a tendency among women to answer more incorrectly compared to the male group. The results from question 26, where students were asked if doctors can legally break the patient’s medical secrecy if the patient gives their consent, show that being a male student multiplies by 4.52 the probability of answering correctly when compared to the female group (95%CI: 1.76–11.60, *p* = 0.002). The same happens in question 29, where the students were presented with a situation in which their neighbor had just been hospitalized in the ward they were rotating in. The students had to decide if they would tell their family/friends about it. The results show that being a male student multiplies by 2.59 the probability of answering correctly when compared to the female group (95% CI: 1.15–5.85, *p* = 0.022).

### 3.5. Association between Academic Year and Knowledge about Ethical Issues

We compared the results between two groups, the first group being 1st to 3rd year students, and the second group being students from subsequent years (4th to 6th). The results regarding this variable are shown in Figure 3. Please note that, although 3rd year students are not pre-clinical students, they had barely rotated when surveyed; for that reason, we have included them in the first group.

Regarding question 25, concerning medical secrecy and situations in which doctors are legally allowed to break it, the results suggest that being a 4th to 6th year student increases 4.15 times the probability of giving the correct answer (*p* = 0.031, CI 1.14–15.08). The same happens in question 39 about uploading a picture to Instagram showing the auscultation of a patient who has a characteristic tattoo, having not asked for his consent, although his face is not shown. In this case, the results suggest that being a 4th to 6th year student increases 7.08 times the probability of giving the correct answer (*p* = 0.02, CI 1.36–36.78). Question 28 deserves special mention, where students were presented with a situation in which a patient prefers them not to be present during the care process. The results suggest that being a 4th to 6th year student increases 2.12 times the probability of answering correctly (*p* = 0.053, CI 0.99–4.52).

In general terms, we found that students in the final years of medical school (4th–6th grade) showed a tendency to respond more correctly than those in the first years (1st–3rd grade).

### 3.6. Association between Parents’ Educational Level and Knowledge

A comparison was made between two groups according to the level of studies reached by the students’ parents. A total of 70 out of 129 students reported that both of their parents are graduates, which means 54% of all the students. The results are shown in Figure 4.

According to the results from question 30, about breaking medical secrecy in case of a disease of public health significance (DOPHS), being a medical student whose parents are graduates multiplies nine-fold the probability of answering correctly compared to the group of non-both graduate parents (OR = 9, 95% CI: 1.05–77.35, *p* = 0.045). The same happens when analyzing the results from question 36, where students were presented with a situation in which an angry patient shouted at them because they had waited for medical attention for a long time. The students had to decide whether to delay medical care or not due to their bad manners. According to our results, being a medical student whose parents are graduates multiplies 2.26 times the probability (odds) of answering correctly (95% CI: 1.09–4.68, *p* = 0.028).

## 4. Discussion

The results of this article suggest that students in Spanish medical schools have some gaps in knowledge or practice regarding patient’s right to intimacy, including, but not limited to, insufficient information/permission from the patient in order to be present in medical procedures, accessing medical records using doctors’ username and password and sharing patient’s images on social media. Older students, men, students in higher years of study and students whose parents’ educational level is higher tend to answer questions regarding patient’s right to intimacy more correctly. However, due to the small sample size of our study, further research would be necessary to corroborate these results.

Our study shows general knowledge among doctors and students regarding the sharing of health-care center internal network usernames and passwords. However, this situation is quite common in hospitals, since more than 70% of medical staff members report having obtained another medical staff member’s password, according to a study [12]. Furthermore, this study reported that this happened more than four times on average [12].

Regarding patients’ right to know that there are students present in their healthcare procedure, most students reported they “always” or “quite often” carry their identification badge when rotating in the hospital, but only a few of them return their ID badge when ending their rotations, despite its return being compulsory. According to our results, wearing these ID cards among students should be enhanced, since patients have the right to be informed of the presence of students during their health-care, and their use has shown patients’ improvement in the identification of health-care personnel and an increase in overall satisfaction [13].

Despite personnel in training having the obligation to sign a confidentiality commitment agreement at the beginning of their time at the health center where they are being trained, our study shows a generalized non-compliance with this obligation during students’ clinical rotations. Moreover,, enormous differences have been shown between the results from 3rd–5th grade students and 6^th^ grade students, suggesting that this commitment to confidentiality is mainly found among medical students in their final year.

Our study also shows a lack of experience with dummies, roleplays, interactive dummies simulating patients, etc. among medical students during their preclinical years, which means that health-care centers or medical schools are not providing enough of these resources. However, the use of interactive dummies, roleplays and other learning techniques among medical students prior to their experience with real patients have been shown to improve students’ communication skills, confidence and teamwork abilities [14,15,16,17,18]. Some of our results suggest a general awareness among students about attitudes that conform to generally accepted standards of courtesy and kindness when dealing with patients, as well as about the obligation to request patients’ consent for students to witness the clinical procedure and situations in which the decision is delegated Regarding the latter, some studies report that patients are sometimes uninformed of students’ presence during their care process [19].

Regarding the influence that the academic year of study has on the student’s knowledge of current legislation, important differences were shown between 1st–3rd grade students and 4th–6th grade students, with the latter group being more precise when facing situations where patients’ right to privacy might be violated. Similar results were obtained when analyzing the influence of age in the students’ perception of these situations. This variable may appear to correlate to “year of study”, since students in higher academic years are usually older. Notwithstanding, sometimes, students’ “year of study” and “age” are not strictly bound for several reasons, such as entering medicine at an older age than usual, having trouble with some subjects or leaving the degree and returning years later. It is noteworthy that medical law teaching is usually taken in the final years of study in Spanish Medical Schools, despite several articles supporting the inclusion of legal education not only in advanced, but also in preclinical years [20,21,22]. However, the differences were not as marked as expected, showing a significant lack of knowledge regarding patients’ privacy even in higher years of study and in older students. In this sense, legal education has been shown to lessen liability risks and inter-professional tensions, in order to help build patient–doctor relationships, and even to improve patient support and health [20]. Thus, more research would be needed to assess whether health policy training in medical schools is adequate, in order to prepare students for their future challenges [23,24].

Although, according to our results, huge differences were shown between male and female students in their perception of clinical situations that may affect patients’ right to privacy, especially regarding personal data protection and confidentiality. Female students’ higher tendency to suffer anxiety and stress effects from contact with patients [25,26] may explain these differences, as well as facing more obstacles during their clinical rotations when compared to males, such as poor mentoring and less support from hostile nurses [27]. However, some other studies report no significant differences between male and female students on their average grades [28], and others even report higher average grades among female students [29]. It is not clear why male students showed higher knowledge rates than females in our study, since the current literature regarding differences in knowledge according to gender is controversial. It is problematic to reach a definite conclusion about the rates of knowledge in men, given the relatively small sample of men analysed, because 75% of the participants in our study were women, which is very similar to the current gender distribution of medical students in Spanish universities. Therefore, more high-quality studies on this are needed in this regard.

Differences according to parents’ educational level were remarkable in aspects related to attitudes that conform to generally accepted standards of courtesy and kindness, to sharing or storing patients’ information and to complying with confidentiality regarding intimate data and biographical aspects of the patient. Several studies report that higher parental socioeconomic status is correlated with better performances and mental health among students [30,31,32]. This association may explain our results, since higher educational level is usually associated with higher socioeconomic status. However, more research on this is needed, in order to determine how parents’ educational level influences medical students.

We conducted a study on a sample of 129 students from various Spanish medical schools. The students exhibited a non-significant degree of misjudgment in certain aspects regarding the patient’s right to privacy. Male and older students’ knowledge of current legislation was higher, as well as that of students whose parents were both university-educated or were in advanced years.

## 5. Conclusions

In conclusion, our study constitutes a first step towards the identification of factors susceptible to compromising the patient’s right to privacy during students’ clinical rotations. In this way, our results suggest that patients’ right to privacy might not be fully guaranteed during students’ clinical rotations. Furthermore, our results also indicate a lack of knowledge regarding current legislation among medical students in general terms. However, given the relatively small sample, more high-quality studies on this are needed in this regard.

## Figures and Tables

**Figure 1 ijerph-19-11067-f001:**
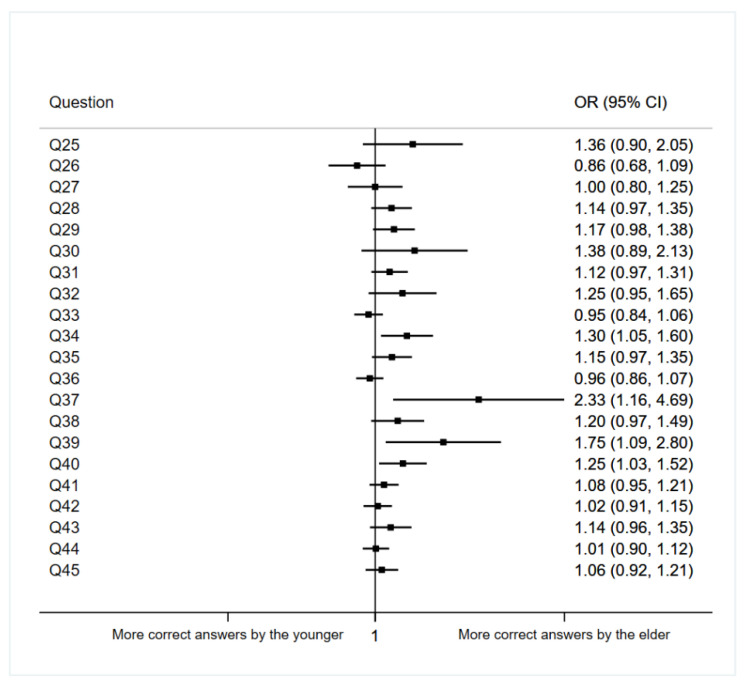
Association between questions regarding students’ opinions and age (per year).

**Figure 2 ijerph-19-11067-f002:**
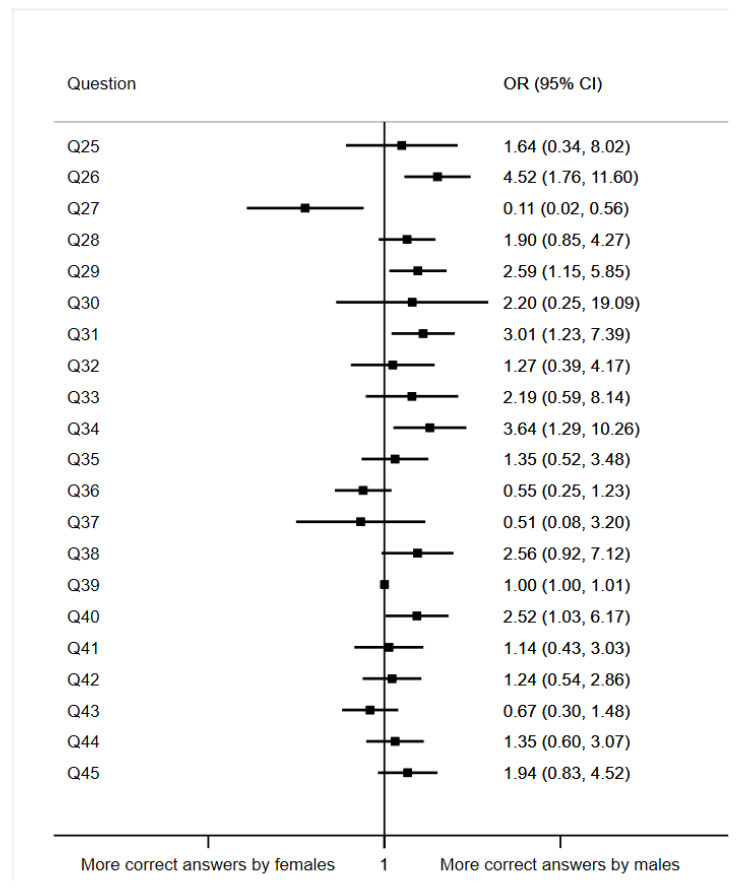
Association between questions regarding students’ opinions and sex (ref. female).

**Figure 3 ijerph-19-11067-f003:**
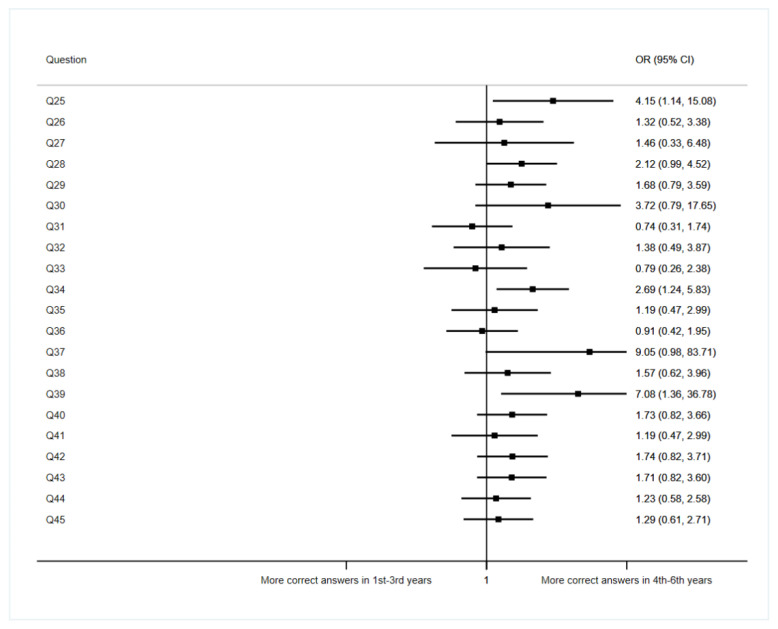
Association between questions regarding students’ opinions and academic year (ref. 1st–3rd.).

**Figure 4 ijerph-19-11067-f004:**
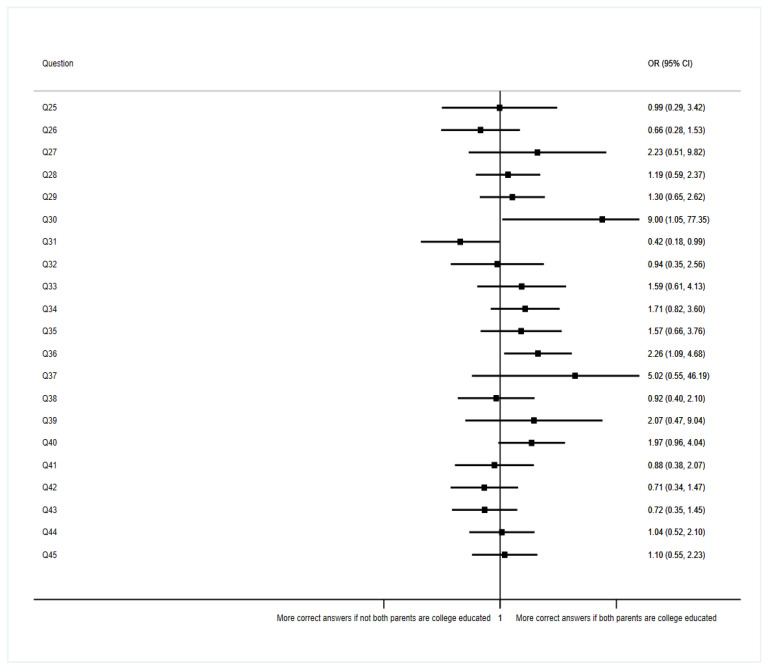
Association between questions regarding students’ opinions and parents’ educational background (ref Not both parents’ college-educated).

**Table 1 ijerph-19-11067-t001:** Main characteristics of the students included in study.

Variable	Category	Total
**Age (years) mean ± standard deviation**		22.1 ± 3.5
**Gender**	Man	33 (25.6)
	Woman	94 (72.9)
	NA	2 (1.5)
**Parents with college education**	Not both parents college-educated	59 (45.7)
	Both parents college-educated	70 (54.3)
**Current academic year**	From 1st to 3rd	42 (32.6)
	From 4th to 6th	87 (67.4)
**Subjects from previous academic years**	No	102 (79)
	Yes	27 (21)
**Current average grade ***	5–6.9	29 (22.5)
	7–8.9	83 (64.3)
	≥9	6 (4.7)
	NA	11 (8.5)
**Number of hospital services the student has rotated at**	0 services	21 (16.3)
	From 1 to 9 services	82 (63.6)
	10 or more services	26 (20.1)

NA: Not available. * Qualifications in Spanish universities are graded from 0 to 10. A grade of 5 is the required qualification to pass any subject.

**Table 2 ijerph-19-11067-t002:** Students’ experience when rotating at the hospital. Only students in 3rd–6th years are included.

Question	Statements	Category	Number of Students from 3rd to 6th Year
Q1	During my clinical rotation, I am constantly supervised and accompanied by my doctor.	Rarely	2 (1.9)
		Sometimes	18 (16.7)
		Quite often	57 (52.8)
		Always	29 (26.9)
		N/A	2 (1.9)
Q2	At the beginning of my rotations, I signed a confidentiality commitment provided by the health center where I carry out the practice.	No	65 (60.2)
		Yes	33 (30.6)
		N/A	10 (9.3)
Q3	During my clinical rotation, the doctor provides me his/her username and password for the hospital’s intranet.	Always	2 (1.9)
		Quite often	1 (0.9)
		Sometimes	4 (3.7)
		Rarely	16 (14.8)
		Never	83 (76.9)
		N/A	2 (1.9)
Q4	When I am at the hospital, I wear the identification card (name and surname, photograph and “student in training”), and carry it in a visible place.	Never	3 (2.8)
		Rarely	12 (11.1)
		Sometimes	6 (5.6)
		Quite often	20 (18.5)
		Always	66 (61.1)
		N/A	1 (0.9)
Q5	The health-care center (not the medical school) takes charge of providing me with a student identification card for the rotation.	No	52 (48.2)
		Yes	52 (48.2)
		N/A	4 (3.7)
Q6	When the academic year is finished, I am obligated to give my student identification card back.	No	86 (79.6)
		Yes	2 (1.9)
		N/A	20 (18.5)
Q7	When assigning the clinical rotations, the faculty informs me of what hospital service I am assigned to. In addition, they also inform me of which doctor will be responsible for my rotation.	Never	1 (0.9)
		Rarely	2 (1.9)
		Sometimes	8 (7.4)
		Quite often	31 (28.7)
		Always	65 (60.2)
		N/A	1 (0.9)
Q8	On the first day of each rotation, my assigned tutor already knew how many students we were and our names, our schedule and timetable, etc.	Never	32 (29.6)
		Rarely	47 (43.5)
		Sometimes	15 (13.9)
		Quite often	12 (11.1)
		Always	1 (0.9)
		N/A	1 (0.9)
Q9	In preclinical years (mainly 1st and 2nd) we have used mannequins, simulated patients and/or roleplays, in order to acquire skills for the “patient-student” relationship in subsequent clinical courses.	No	46 (42.6)
		Yes	58 (53.7)
		N/A	4 (3.7)
Q10	I have got nervous on some occasion during a health care process in my rotation, and I have missed not having practiced before with a mannequin, simulated patient, using roleplays, etc.	Yes	81 (75)
		No	25 (23.2)
		N/A	2 (1.9)
Q11	During my rotation, I have access to the patient’s medical history and I could have modified it.	Always	11 (10.2)
		Quite often	20 (18.5)
		Sometimes	25 (23.2)
		Rarely	26 (24.1)
		Never	25 (23.2)
		N/A	1 (0.9)
Q12	I do clinical interviews with patients and read their medical reports without the presence of my responsible doctor.	Always	3 (2.8)
		Quite often	28 (25.9)
		Sometimes	44 (40.7)
		Rarely	22 (20.4)
		Never	11 (10.2)
		N/A	0 (0.0)
Q13	During my rotation, the doctor knocks at the door and greets the patients when entering their rooms.	Never	1 (0.9)
		Rarely	2 (1.9)
		Sometimes	8 (7.4)
		Quite often	47 (43.5)
		Always	49 (45.4)
		N/A	1 (0.9)
Q14	More than 3 students have been present at the same time in some health-care process of a patient.	Always	2 (1.9)
		Quite often	29 (26.9)
		Sometimes	49 (45.4)
		Rarely	16 (14.8)
		Never	12 (11.1)
		N/A	0 (0.0)
Q15	I have sent a patient’s medical history to my personal email, without expressly asking the patient for permission.	Yes	3 (2.8)
		No	104 (96.3)
		N/A	1 (0.9)
Q16	I have used my personal mobile phone during a patient’s care process for things unrelated to learning (WhatsApp, social networks, etc.)	Quite often	2 (1.9)
		Sometimes	18 (16.7)
		Rarely	33 (30.6)
		Never	55 (50.9)
		N/A	0 (0.0)
Q17	My assigned doctor informs me about patients’ rights and corrects me in case of not acting correctly on issues of personal treatment, ethics, privacy, dignity, etc.	Never	15 (13.9)
		Rarely	24 (22.2)
		Sometimes	28 (25.9)
		Quite often	20 (18.5)
		Always	12 (11.1)
		N/A	9 (8.3)
Q18	Before a patient’s care process, my assigned doctor informs him/her (or his/her representative) of the presence of students and asks him/her for verbal consent for the students to witness the clinical act.	Never	4 (3.7)
		Rarely	25 (23.2)
		Sometimes	33 (30.6)
		Quite often	34 (31.5)
		Always	12 (11.1)
		N/A	0 (0.0)
Q19	If the doctor considers it appropriate for me to carry out some type of exploration or procedure, he/she again asks for the verbal consent of the patient (or his/her representative).	Never	3 (2.8)
		Rarely	16 (14.8)
		Sometimes	26 (24.1)
		Quite often	37 (34.3)
		Always	26 (24.1)
Q20	Some doctor told me private aspects about a patient without any clinical interest (e.g., “she is the mother of the mayor”, “she is the sister of the hospital manager”, “he likes to go to certain places”, etc.).	Yes	63 (58.3)
		No	38 (35.2)
		N/A	7 (6.5)
Q21	The management of the health centers in which I carry out my training rotations (medical director, hospital manager, etc.) are responsible for explaining the ethical principles and basic rules of action in the patient care process.	Never	55 (50.9)
		Rarely	25 (23.2)
		Sometimes	16 (14.8)
		Quite often	5 (4.6)
		Always	3 (2.8)
		N/A	4 (3.7)
Q22	During my rotations, the doctor introduces and identifies him/herself when entering the patients’ rooms, as well as introduces me and identifies me as a student.	Never	4 (3.7)
		Rarely	17 (15.7)
		Sometimes	35 (32.4)
		Quite often	39 (36.1)
		Always	12 (11.1)
		N/A	1 (0.9)
Q23	During my rotations, I have access to patients’ identification data, such as name, surname, age, address, etc. (I see it in medical record or report, my assigned doctor tells me, etc.), not only purely clinical data.	Always	38 (35.2)
		Quite often	38 (35.2)
		Sometimes	25 (23.2)
		Rarely	4 (3.7)
		Never	2 (1.9)
		N/A	1 (0.9)
Q24	The health-care centers in which I carry out my rotations have systems that allow students to read clinical reports about patients without being able to see their identification data (name, address, profession, etc.).	No	66 (61.1)
		Yes	11 (10.2)
		N/A	31 (28.7)

**Table 3 ijerph-19-11067-t003:** Students’ opinions on different situations.

Question	Statements	Category	Total	Current Academic Year		*p*
				From 1st to 3rd	From 4th to 6th	
Q25	Being a student and not a doctor, I am legally not compelled to keep medical secrecy.	Incorrect answer	11	7 (63.64)	4 (36.36)	0.021
		Correct answer	118	35 (29.66)	83 (70.34)	
Q26	If a patient gives his/her consent, the doctor is legally allowed to “break” medical secrecy.	Incorrect answer	77	25 (32.47)	52 (67.53)	0.556
		Correct answer	30	8 (26.67)	22 (73.33)	
		N/A	22	9 (40.91)	13 (59.09)	
Q27	If a patient dies, medical secrecy disappears with him/her.	Incorrect answer	8	3 (37.50)	5 (62.50)	0.063
		Correct answer	110	32 (29.09)	78 (70.91)	
		N/A	11	7 (63.64)	4 (36.36)	
Q28	During my rotation in the hospital, a patient wants me not to be in his care process because I am a student. I refuse, since it is a “university hospital”.	Incorrect answer	67	27 (40.30)	40 (59.70)	0.051
		Correct answer	62	15 (24.19)	47 (75.81)	
Q29	During your rotation in cardiology, a neighbor of yours gets hospitalized in your wards. When you get home, you tell your parents/friends about it.	Incorrect answer	72	27 (37.50)	45 (62.50)	0.178
		Correct answer	57	15 (26.32)	42 (73.68)	
Q30	If a doctor diagnoses a patient with a disease of public health significance (for example, cholera), the doctor is legally allowed to “break” medical secrecy and report to health authorities.	Incorrect answer	7	4 (57.14)	3 (42.86)	0.001
		Correct answer	110	29 (26.36)	81 (73.64)	
		N/A	12	9 (75.00)	3 (25.00)	
Q31	I take a photograph of a patient’s admission report, which would be useful for preparing some assignment. I will not share it with anyone, I will simply store it in the phone’s photo gallery.	Incorrect answer	100	31 (31.00)	69 (69.00)	0.483
		Correct answer	29	11 (37.93)	18 (62.07)	
Q32	During your rotation in psychiatry, a patient suspected of domestic violence addresses you because he wants some medication to calm his headache. You are alone. You ignore it and don’t tell your doctor.	Incorrect answer	18	7 (38.89)	11 (61.11)	0.537
		Correct answer	111	35 (31.53)	76 (68.47)	
Q33	If a doctor diagnoses a patient with a severe contagious disease, the doctor is legally allowed to “break” medical secrecy and tell the patient’s partner, even if the patient does not want it to be told, in order to avoid potential damage of the partner’s health.	Incorrect answer	21	5 (23.81)	16 (76.19)	0.016
		Correct answer	88	25 (28.41)	63 (71.59)	
		N/A	20	12 (60.00)	8 (40.00)	
Q34	During my rotation, I see a patient with Fournier’s gangrene. This disease is not seen every day, so I send a WhatsApp group with class friends a photo of the perineal necrosis, without informing the patient, but without giving information about his identity.	Incorrect answer	42	20 (47.62)	22 (52.38)	0.011
		Correct answer	87	22 (25.29)	65 (74.71)	
Q35	In case of a patient who is expected to stay in hospital more than 15 days, if the doctor has already asked the patient for his verbal consent for me to be present at the clinical events and even perform physical examinations, it is not necessary to ask again the patient for his verbal consent during the rest of his stay.	Incorrect answer	102	34 (33.33)	68 (66.67)	0.715
		Correct answer	27	8 (29.63)	19 (70.37)	
Q36	During your emergency room rotation, a patient is continuously shouting and insulting the staff because he has been waiting to be attended for quite some time. For this reason, you attend before to other patients who have arrived later than him.	Incorrect answer	48	15 (31.25)	33 (68.75)	0.807
		Correct answer	81	27 (33.33)	54 (66.67)	
Q37	During your rotation through the internal medicine wards, you perform a physical examination of a terminally ill patient. After finishing, you go with your doctor to the wards to continue visiting patients. However, when you leave the patient’s room you realize that you have exposed the patient’s genitalia and you tell the doctor. The patient has dementia and has no family/friends who could complain. The doctor tells you “it is not necessary; nobody comes to see him”.	Incorrect answer	5	4 (80.00)	1 (20.00)	0.021
		Correct answer	124	38 (30.65)	86 (69.35)	
Q38	If a judge requests a doctor to testify at a trial, the doctor is legally allowed to “break” medical secrecy regarding that patient.	Incorrect answer	36	11 (30.56)	25 (69.44)	0.002
		Correct answer	64	14 (21.88)	50 (78.13)	
		N/A	29	17 (58.62)	12 (41.38)	
Q39	I upload to Instagram a photo in which I am auscultating a patient. I do not expressly ask the patient for permission. The patient has a characteristic tattoo on the sternum area (visible in the photo), but his face cannot be seen.	Incorrect answer	8	6 (75.00)	2 (25.00)	0.008
		Correct answer	121	36 (29.75)	85 (70.25)	
Q40	I upload to Instagram a photo in which I am auscultating a patient. I do not expressly ask the patient for permission, but her face cannot be seen, so there is no way to know her identity.	Incorrect answer	50	20 (40.00)	30 (60.00)	0.151
		Correct answer	79	22 (27.85)	57 (72.15)	
Q41	During your rotation in traumatology, the surgeon photographs with her personal mobile phone anatomic areas of patients with large cosmetic defects. After the operation, she photographs the area again to make a comparison. In order to know which patient the image belongs to, she labels images with medical record numbers.	Incorrect answer	102	34 (33.33)	68 (66.67)	0.715
		Correct answer	27	8 (29.63)	19 (70.37)	
Q42	In case of an under-18 patient, the doctor must ask the patient’s legal representative for verbal consent about my presence as a student in the health care process.	Incorrect answer	47	19 (40.43)	28 (59.57)	0.149
		Correct answer	82	23 (28.05)	59 (71.95)	
Q43	In the previous case (under-18 patient), once verbal consent has been asked, the legal representative decides on his/her own, and has no obligation to listen to what the minor thinks about what has been reported.	Incorrect answer	56	22 (39.29)	34 (60.71)	0.153
		Correct answer	73	20 (27.40)	53 (72.60)	
Q44	In case of a patient with limited decision-making capacity, the doctor must ask the family/partner/legal representative of the patient for verbal consent about my presence as a student in the care process.	Incorrect answer	54	19 (35.19)	35 (64.81)	0.589
		Correct answer	75	23 (30.67)	52 (69.33)	
Q45	In case of an incapacitated patient (with a judicial sentence), the doctor must ask the family/partner/legal representative of the patient for verbal consent about my presence as a student in the care process.	Incorrect answer	53	19 (35.85)	34 (64.15)	0.505
		Correct answer	76	23 (30.26)	53 (69.74)	

N/A: Not available.

## Data Availability

The data presented in this study are available on request from the corresponding author. All data generated or analysed during this study are included in this published article and its additional files.

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
