# Peer review of "Evolution of Medical Students’ Perception of the Patient’s Right to Privacy"

_ijerph, 2022, doi:10.3390/ijerph191711067_

Round 1

Reviewer 1 Report

This paper analysed the medical students' perception of patients' privacy with questionnaires. However, there are some weakness of this work:

1. Only 22 participants are not enough to have the conclusion of "Spanish hospitals and health-care centers should work on the promotion of informative courses regarding current legislation 331 among medical staff members and medical students. " The authors also did not mention whether the 22 participants are from the same hospital or different hospitals.

2. The gender distribution is imbalanced in the 22 participants.

Author Response

This paper analysed the medical students' perception of patients' privacy with questionnaires. However, there are some weakness of this work:

Only 22 participants are not enough to have the conclusion of "Spanish hospitals and health-care centers should work on the promotion of informative courses regarding current legislation 331 among medical staff members and medical students. " The authors also did not mention whether the 22 participants are from the same hospital or different hospitals.

The reviewer is right. It is true that with only 33 men from different universities, the results cannot be generalized to the Spanish population. We have highlighted our limitation derived from the small sample size in several sections of the articles and we have also eliminated our generalization of the results.

The gender distribution is imbalanced in the 22 participants.

About 75% of the participants in our study were women, which is very similar to the current gender distribution of medical students in Spanish universities. Although according to our results, male students showed higher rates of knowledge than female students, given that the current literature on differences in knowledge according to gender is controversial, it is problematic to reach a definite conclusion about the rates of knowledge in men, given the relatively small sample of men analysed, so more high-quality studies are needed in this regard.

This limitation has been incorporated into the manuscript

Reviewer 2 Report

The paper analyzes answers of 129 medical students in Spain regarding privacy issues of patient. Altogether, the article is fluently written. Moreover, the findings are interesting and even quite worryingly, meaning it is good that the study was conducted so that hopefully the missing knowledge of some students regarding patient privacy can be addressed. However, my concerns relate to the theory underlying the study and the generalizations that can be drawn from the study. There is not much theory on how the questions were developed. Did the authors come up with the questions themselves? On what grounds were they based? Please elaborate.   The logistic regression model should be explained in more detail. What were all the explanatory and dependent variables? Please add also the corresponding equation.   Older students, men, students in higher years of study and students whose parents’ educational level is higher tend to answer questions regarding patient’s right to intimacy more correctly -> make sure to emphasize that this is the result only of your small limited data, cannot be generalized In general, the limitations arising from the very small data should be highlighted at several places.   question Q29 is missing. Although it would be especially interesting since it was more correclty answered by the younger   Please note that, although 3rd year students 207 are not pre-clinical students, they had barely rotated when surveyed -> serial stop missing at the end of the sentence     Very often, a space is missing between two words. Here are just some of numerous examples: correlateto communicationskills studentsfor evento

->correct all

Author Response

The paper analyzes answers of 129 medical students in Spain regarding privacy issues of patient. Altogether, the article is fluently written. Moreover, the findings are interesting and even quite worryingly, meaning it is good that the study was conducted so that hopefully the missing knowledge of some students regarding patient privacy can be addressed. However, my concerns relate to the theory underlying the study and the generalizations that can be drawn from the study.

There is not much theory on how the questions were developed. Did the authors come up with the questions themselves? On what grounds were they based? Please elaborate.  

We analyzed the level of compliance of the SSI/81/2017 Spanish Order during medical students’ rotations. This order regulates the right to privacy owned by every patient during their care processes and addresses health-care staff obligations concerning this. We also analyzed the level of knowledge regarding this legislation among medical students.

This information is attached as supplementary material and in the method section.

The logistic regression model should be explained in more detail. What were all the explanatory and dependent variables? Please add also the corresponding equation.  

In the method section, the logistic regression model has been explained in more detail, adding the corresponding equations

Older students, men, students in higher years of study and students whose parents’ educational level is higher tend to answer questions regarding patient’s right to intimacy more correctly -> make sure to emphasize that this is the result only of your small limited data, cannot be generalized In general, the limitations arising from the very small data should be highlighted at several places.  

The reviewer is right. We have highlighted our limitation derived from the small sample size in several sections of the articles.

question Q29 is missing. Although it would be especially interesting since it was more correclty answered by the younger  

This question was removed from the questionnaire because it did not allow analysis of the level of compliance with any item of the SSI/81/2017 Spanish. In the new version of the article the questions have been renamed. Now question 29 is question 30 from the previous version.

Please note that, although 3rd year students 207 are not pre-clinical students, they had barely rotated when surveyed -> serial stop missing at the end of the sentence    

Now the sentence has been finished as follows:

“We compared the results between two groups, the first group being 1st to 3rd year students, and the second group students from subsequent years (4th to 6th). Results regarding this variable are shown in Figure 1c. Please note that, although 3rd year students are not pre-clinical students, they had barely rotated when surveyed, for that reason we have included them in the first group”.

Very often, a space is missing between two words. Here are just some of numerous examples: correlateto communicationskills studentsfor evento

 These typographical errors have been corrected

Round 2

Reviewer 2 Report

The authors addressed all comments. I think this is an interesting article that can be published in its current form.